# Changes in plantar load distribution in legally blind subjects

Ketlin Jaquelline Santana Castro[1], Railson Cruz Salomão[2], Newton Quintino Feitosa, Jr.[2], Leonardo Dutra Henriques[3], Ana Francisca Rozin Kleiner[4], Anderson Belgamo[5], André Santos Cabral[6], Anselmo Athayde Costa e Silva[7], Bianca Callegari[7,8], Givago Silva Souza[1,2]*

1 Instituto de Ciências Biológicas, Universidade Federal do Pará, Belém, Brazil, 2 Núcleo de Medicina Tropical, Universidade Federal do Pará, Belém, Brazil, 3 Instituto de Psicologia, Universidade de São Paulo, São Paulo, Brazil, 4 Departamento de Fisioterapia, Universidade Federal de São Carlos, São Carlos, Brazil, 5 Instituto Federal de São Paulo, Piracicaba, Brazil, 6 Centro de Ciências Biológicas e da Saúde, Universidade do Estado do Pará, Belém, Brazil, 7 Master's Program in Human Movement Sciences, Federal University of Pará, Belém, Pará, Brazil, 8 Laboratory of Human Motricity Sciences, Federal University of Pará, Belém, Pará, Brazil

* givagosouza@ufpa.br

**Data Availability Statement:** All relevant data are within the manuscript and its Supporting Information files.

**Funding:** This study was supported by: APESPA, 2019/589349, Mr Anselmo Athayde Costa e Silva

## Abstract

We investigated the impact of visual impairment on balance control. We measured the center of pressure (COP) between the two feet and plantar surface pressures on each foot in 18 normal-sighted participants and compared their data with measures from 18 legally blind participants, either acquired or congenital. Pressures were measured in open- and closed-eye conditions using a baropodometric resistive plate. In the eyes-open condition, there were no differences between the sighted and legally blind groups in COP displacement. However, participants with visual loss had significantly increased pressures in two metatarsal regions (M1 and M2 zones) of the plantar surface in both viewing conditions ($p < 0.05$). The differences in pressure measures between the normally sighted and legally blind groups could be attributed mainly to the subgroup of subjects with acquired impairment. Our findings suggest that subjects with visual impairment present increased metatarsal pressures (i.e. forefoot), not yet associated to anterior displacement of COP or impaired balance control.

## Introduction

Dynamic maintenance of balance while standing in humans relies on information from visual, vestibular, and proprioceptive inputs required by the brain to appropriately generate the complex array of motor commands needed to achieve equilibrium in a standing position [1–4]. Sensory impairment can impede adaptive postural control mechanisms and lead to equilibrium loss (i.e., visual [5–7]; vestibular [8, 9]; proprioceptive [10, 11]; vestibular and proprioceptive [12]).

The contribution of the visual inputs to the balance control is a hot topic and have been previously investigated. Maintenance of balance control in conditions of visual loss is aided by

Conselho Nacional de Desenvolvimento Científico e Tecnológico (BR), 431748/2016-0, Mr Givago Silva Souza Conselho Nacional de Desenvolvimento Científico e Tecnológico, 310845/2018-1, Mr Givago Silva Souza. The funders had no role in study design, data collection and analysis, decision to publish, or preparation of the manuscript.

vestibular and proprioceptive inputs and is manifested via compensatory adjustments of postural weighting [13, 14]. Since postural changes in standing position reflect modifications in our body weight load on the plantar surface, measurement of plantar pressure can be used to quantify the influence of visual input on posture control [15]. A variety of tools have been employed to quantify balance control, such as stabilometry, dynamometry, video system analysis, electromyography during the execution of quiet stance, tandem Romberg test, one leg stance, and reaction-time tasks [13, 14, 16–23].

Measuring postural control in subjects with sensory impairment can quantify the effects of the impairment on balance control mechanisms [6–12, 16, 23]. There are mixed findings regarding the impact of visual loss on balance control and equilibrium [6–12, 24–26]. Some research finds that people with visual deficits have impaired balance control [16, 17], while other research finds minor or no differences in static and dynamic postural control between sighted and visually impaired subjects [13, 18, 19, 27]. Subjects with altered binocular vision have been found to have significantly altered measures of foot plantar pressures, and blind subjects can have prolonged foot-to-ground contact during gait [15, 24].

One variable that could affect balance control is whether the visual function loss is congenital or acquired. Previous studies investigated balance control in subjects with congenitally and acquired blindness [28, 29] and reported no differences between control and congenitally blind individuals, but participants with acquired blindness were less stable than controls. Some studies suggest that individuals with congenital visual impairment develop effective somatosensory and vestibular mechanisms to compensate for a lack of visual information since the birth [13, 14].

In the present study we aimed to measure balance control by direct and indirect (plantar surface pressure distribution) measures of COP from congenitally and acquired blind subjects and compare these data with data from sighted subjects.

Based on prior data, we hypothesized that subjects with acquired visual impairment would be more susceptible to disturbances in balance control, and hence maintenance of normal COP, and that associated deficits would be found in the load of pressures on the plantar surface of the foot. However, it is not obvious which specific regions of an individual foot would manifest changes in balance control.

## Methods

### Ethical consideration

All procedures were approved by the Ethical Committee for Research in Humans of the Science Health Institute of the Federal University of Pará (report #3.040.281/2018) and followed the STROBE statement. Written informed consent was obtained from all the participants before the procedures start. The visually impaired subjects read a Braille version of the instructions document or were verbally instructed before giving consent. Data were acquired between December 2017 to September 2019.

### Subjects

Our sample consisted of 36 subjects between 18 and 50 years-old (18 sighted participants, 18 visually impaired participants). The sample of legally blind subjects was recruited from the José Alvares de Azevedo school for the blind and visually impaired. Participants were not compensated financially. Inclusion criteria were impairment of the visual perception and no motor function disturbances to keep an erect posture. Exclusion criteria for both groups were somesthetic, orthopedic, or vestibular, and neurological pathologies, motor disturbances or attention and/or memory deficits.

**Table 1. Description of the visual deficits of the visually impaired participants.**

| Patient | Diagnosis | Blindness | Visual acuity |
|---|---|---|---|
| P1 | Glaucoma | Congenital | LP/LP |
| P2 | Optic nerve atrophy | Acquired (10 years ago) | LP/LP |
| P3 | Glaucoma | Congenital | HM 1.7'/CF 1.7' |
| P4 | Pituitary adenoma | Acquired (17 years ago) | NLP/NLP |
| P5 | Cataract | Acquired (5 years ago) | LP/LP |
| P6 | ND | Congenital | NLP/NLP |
| P7 | ON atrophy | Acquired (13 years ago) | LP/LP |
| P8 | Uveitis | Congenital | NLP/NLP |
| P9 | Cataract | Congenital | LP/LP |
| P10 | Cataract | Acquired (2 years ago) | LP/LP |
| P11 | Cataract | Congenital | LP/LP |
| P12 | Chorioretinitis | Acquired (30 years ago) | 20/200/CF 3.5' |
| P13 | ND | Congenital | LP/LP |
| P14 | Retinitis pigmentosa | Congenital | LP/LP |
| P15 | Glaucoma | Congenital | LP/LP |
| P16 | Glaucoma | Acquired (8 years ago) | NLP/NLP |
| P17 | Cataract | Congenital | LP/LP |
| P18 | Glaucoma | Acquired (3 years ago) | LP/LP |

ND: not diagnosed; ON: Optic neuritis; LP: Light perception; NLP: No light perception; CF: Counting fingers; HM: Hand motion. The visual acuity descriptions for counting fingers or hand motion are expressed as the maximum distance, in feet, from which detection was successful.

An ophthalmologist evaluated all blind participants. We used an ETDRS chart (Xenonio, Brazil) to estimate the visual acuity. All sighted participants had normal or best-corrected visual acuity at 20/20. The visual acuity was recorded in Snellen fraction, and in the cases of very low vision (worse than 20/200), the visual acuity was classified using a semi-quantitative scale: counting fingers (CF), hand motion (HM), light perception (LP), and no light perception (NLP). After the ophthalmological examination and the history of the present illness we divided the sample into acquired and congenital blind participants. All blind subjects that participated in the present study were legally blind (i.e., visual acuity was equal to or worse than 20/200) (Table 1). The definition of blindness is based on foveal vision (central vision), and most of the blind participants had some luminance perception, indicating some peripheral visual function. It is unclear the role of central or peripheral vision in balance control, if both have equal importance, or if both have complementary functionality [30–34].

Ten out of 18 visually impaired participants had congenital visual impairment, while 8 participants had acquired visual impairment. Four of the 18 blind participants had no light perception, while the others had visual acuities ranging from light perception to 20/200 visual acuity (Table 1).

## Physical examination

Age, height, and weight information from all participants were collected. Table 2 describes the physical characteristics of the participants from both groups. All participants carried out a physical evaluation comprising a manual muscle testing on a five-point scale (0 –no muscle strength, 5 –maintain the position when a maximum resistance is applied), assessment for muscle tone, evaluation of superficial and deep reflexes, tactile sensitivity using a brush of the reflex hammer, vibration sense testing with a 128-Hz tuning fork, motor coordination

**Table 2. Physical characteristics of the groups.**

| Variable | CG | VIG | p-value |
|---|---|---|---|
| Age (years) | 31.8 ± 8.3 | 31.5 ± 9.4 | 0.90 |
| BMI (kg/cm$^2$) | 25.4 ± 4.4 | 24.8 ± 5.1 | 0.73 |
| Male/Female | 12 M/ 6 F | 13 M/ 5 F | 0.99 |

Values are present as means and standard deviations for age and BMI.

CG: control group; VIG: visually impaired group; BMI: body mass index.

evaluation using finger-to-nose test and motion of the heel over the shin test. For the physical examination test, all the participants had normal results.

Table 1 describes the physical features of the participants. Both groups were age-, body mass index- (BMI), and male/female proportion matched. The sample was homogeneous to age, BMI, and male/female proportion.

## Apparatus and experimental procedures

All participants stood in normal quiet stance on a baropodometric resistive plate (EPS R-1 model, Loran Engineering, Italy) with 2224 sensors distributed over 48 cm2, with a pressure-range capacity of 50–350 kilo-Pascals (kPa), and a data-acquisition rate of 50 Hz. The individuals were barefoot, with feet held apart at a distance between the shoulders and arms lying along the lateral torso. Sighted participants were asked to direct their gaze to a circular target on the wall a 1 m distance. Visually impaired participants were requested to direct their gaze forward while standing 1 m away from the wall. Conversation was not allowed during the recording sessions, except for the orientations of the participants by the examiners. Simultaneous data acquisition of the center of pressure (COP) displacements and barefoot plantar pressures were performed using Biomech Studio software (Loran Engineering, Italy). Measurements were carried out in open and closed eye conditions, during three trials of 60 seconds in each condition, with 60 seconds of rest between open- and closed-eye conditions. COP displacements along the anteroposterior and mediolateral directions were exported and were used to quantify the postural stability using the parameters of total displacement (COP$_{distance}$) and the area of displacement ellipse (COP$_{area}$) enclosed by the statokinesiogram [35].

Pressure (in kPa) measures from 10 zones of the plantar surface delimited by the Biomech Studio software (forefoot: T1 –zone of the first toe, hallux, T2-5 –zone between the second and fifth toes, M1, M2, M3, M4, and M5 zones–zone of the first, second, third, fourth, and fifth metatarsal heads, respectively; midfoot: MF zones; hindfoot: LH–lateral heel zone and MH–medial heel zone) were quantified as mean (P$_{mean}$), and maximum (P$_{max}$) pressures obtained from three-time series [36]. A schematic representation for analyzing the indirect measure of the postural control based on plantar pressures is shown in Fig 1.

## Data analysis

GraphPad Prism 8 (GraphPad software, Inc., CA, USA) was used to the data analysis. Normality of data was tested using the Shapiro-Wilk test. Non-normal distribution data from control and visually impaired groups were compared using the non-parametric Mann-Whitney U test. We also divided the legally blind group into congenital and acquired subgroups and compared their data to the control using the Kruskal-Wallis test followed by the Dunn post hoc test. We calculated the adjusted p-values for multiplicity using the Bonferroni correction. A Chi-square with Yates' correction was used to compare the proportions of male and female participants of

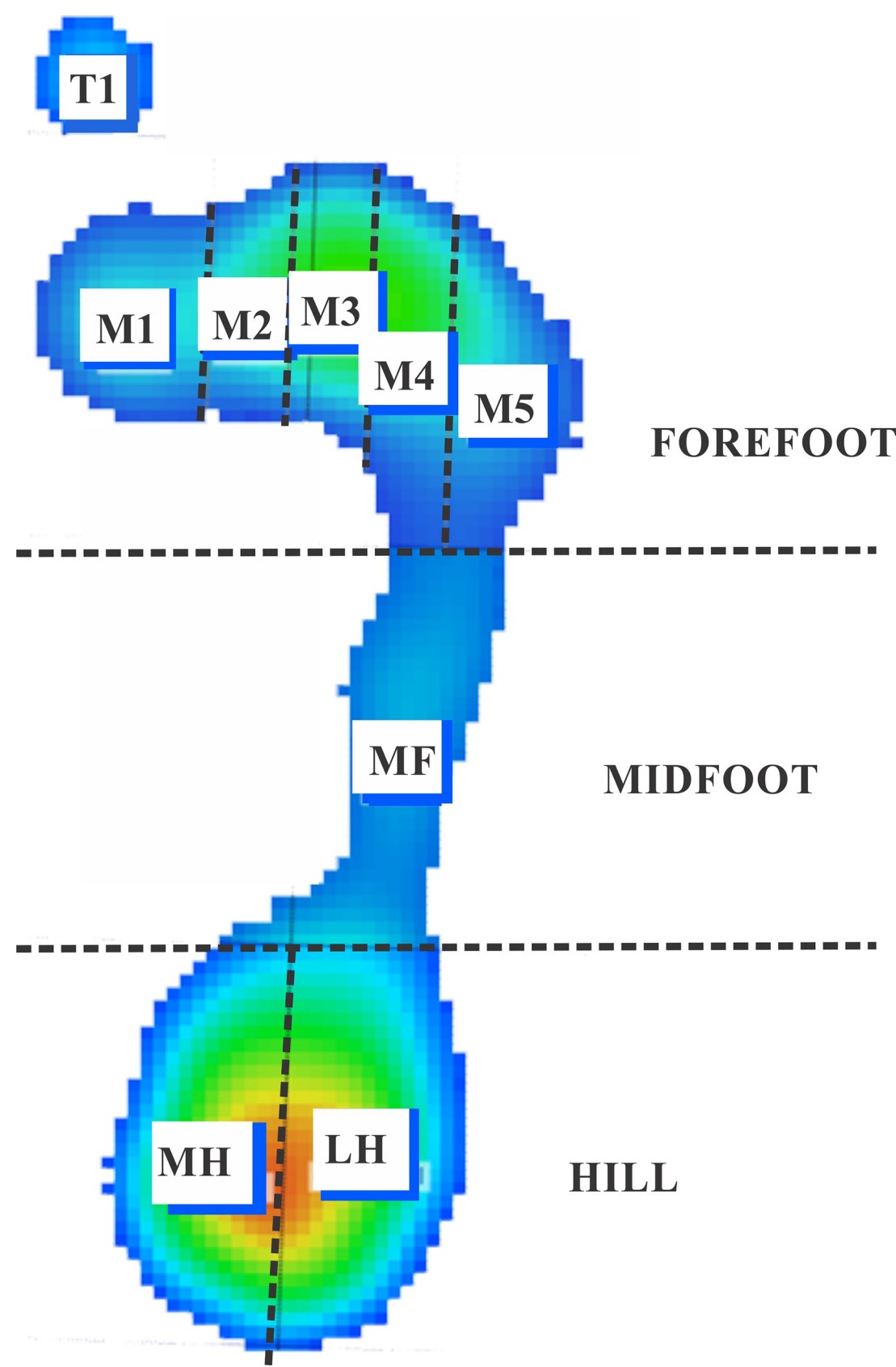

**Fig 1. Representation of the zones of the foot where plantar pressures were quantified.** Forefoot is represented by the T1 (first toe), T2-5 (2nd-5th toes), M1 (1st metacarpal), M2 (2nd metacarpal), M3 (3rd metacarpal), M4 (4th metacarpal), M5 (5th metacarpal) zones; Midfoot is represented by the MF zone; Hindfoot is represented by the medial (MH) and lateral zones (LH).

each group, and Student's t-test was used to compare age, height, weight, and body mass index between groups. A significance level of 0.05 was considered for all the statistical procedures.

## Results

### Balance control comparisons

Typical statokinesiograms were found for all participants, in which anteriorposterior displacements are larger than the laterolateral displacements during the test duration. Fig 2 shows representative statokinesiograms obtained from one representative participant in each group, and no systematic or qualitative differences between the groups could be observed between the statokinesiograms of these participants. Table 3 presents the descriptive statistics for stabilometric variables in the different test conditions for control and visually impaired groups. Both groups' balance control was similar as no significant difference was found between the sighted and the visually impaired groups for $COP_{distance}$ and $COP_{area}$ in either open- and closed-eye conditions. Similarly, no significant difference for stabilometric parameters was found when comparing control, congenital, and acquired visually impaired subjects (Table 4).

### Plantar pressures comparisons

Fig 3 shows an example of a heatmap of the $P_{mean}$ of the feet from the same representative participants shown in Fig 2. In the control participant (Fig 2A), we observed the distribution of pressures occurring mainly on the hallux and mid-to-lateral metatarsus and heel regions. In contrast, in the visually impaired participant, the pressures were mainly localized on between the hallux, metatarsal (M1-M5) region of the left foot, and the heel.

Table 5 compares the plantar pressures between sighted and visually impaired subjects for open- and closed-eye conditions. Blind participants had significantly higher $P_{mean}$ and $P_{max}$ in the M1 and M2 zones ($p < 0.05$), respectively. No significant differences were found among the plantar pressures recorded from sighted, congenital, and acquired visually impaired participants in open-eye conditions. However, in the closed-eye condition, we found that acquired visually impaired participants had significantly higher $P_{mean}$ and $P_{max}$ plantar pressures than the controls in M1 and M2 zones, respectively (Table 6). No significant difference was found between controls and congenital visually impaired participants or between the two visual impairment groups.

## Discussion

This study compared stabilometric and baropodometric measurements between sighted subjects and with subjects with acquired and congenital blindness. Similar to prior investigations, we observed no significant difference of the stabilometric variables ($COP_{distance}$, $COP_{area}$) across the groups, indicating the action of compensatory mechanisms in the balance control of the participants with visual impairment [27, 37].

Additionally, we also observed that most of the foot zones of the legally blind participants had no significant pressure differences compared to controls, except for the higher maximum pressures in the first and second metatarsal heads (M1 and M2 zones) in patients in the closed-eye condition. These differences could be attributed to the contribution of the acquired blind participants' data [15]. The small number of significant differences indicates that mild

## Test condition

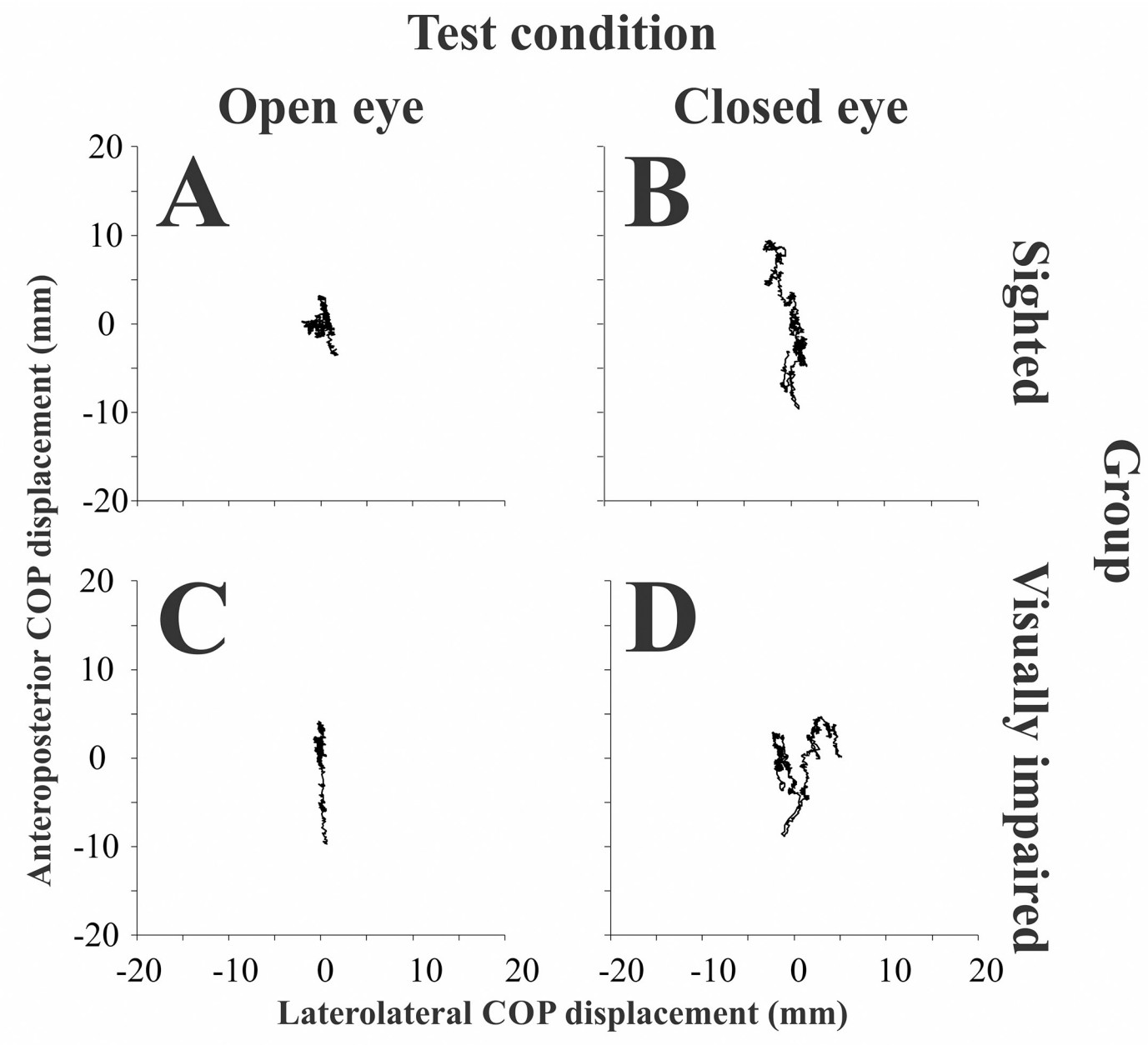

**Fig 2. Statokinesiogram of one representative participant from the sighted group and one representative participant from the blind group.** Recordings from the sighted participant in the open-eye (A) and closed-eye condition (B), and from a visually impaired (blind) participant in the open-eye (C) and closed-eye (D) conditions.

disturbances in the motor control of the balance are present and maybe can be ignored in conventional evaluation of the balance.

Because of the similarity in COP measures obtained from participants with congenital or acquired blindness, a working hypothesis could be that the mean duration of acquired impairment (11 ± 9.2 years) was adequate for the development and consolidation of compensatory balance-control mechanisms.

However, against such a hypothesis, the presence of significant differences in metatarsal head pressures argues for an incomplete establishment of these mechanisms in the participants

**Table 3. Comparison of stabilometric data obtained from control and visually impaired groups.**

| Variable | CG | LBG | P |
|---|---|---|---|
| *Open eye condition* | | | |
| $COP_{distance}$ | 327.46 (93) | 311.23 (89.2) | 0.99 |
| $COP_{area}$ | 87.69 (72.6) | 68.33 (61.9) | 0.86 |
| *Closed eye condition* | | | |
| $COP_{distance}$ | 351.3 (48.3) | 319.75 (101.9) | 0.42 |
| $COP_{area}$ | 110.64 (109.1) | 73.66 (58.8) | 0.5 |

CG: control group; LBG: legally blind group.

with acquired blindness. Congenitally blind subjects would have an opportunity to develop compensatory mechanisms during their development in early childhood, while the ability to develop compensatory mechanisms in acquired visual loss would depend on many factors, including age of onset relative to any possible critical periods, and duration of the visual impairment. Moreover, it is possible that qualitatively different compensatory mechanisms for balance control might be utilized in acquired *vs* congenital visual impairment.

Because we found that differences between the acquired and congenital blind groups in plantar pressures occurred mainly in the closed-eye condition, residual light perception (present in most of the acquired blind group) is a likely an important factor that assists in performance during the balance control task.

In addition, the visual system has more than one pathway to the brain; and conscious visual perception is generated in only one of these pathways (from retina to primary visual cortex, V1). However, other visual pathways could be contributing to the balance control even in visually impaired or blind subjects (for example from retina to superior colliculus or pulvinar nucleus), as has been observed in cases of so-called "blindsight" demonstrating residual vision in several visual diseases [38–40].

We found that subjects in the acquired blind group exhibited an anterior displacement of pressures in the metatarsal zones of the forefoot in order to maintain equilibrium. Such anterior pressure displacements can lead to reflex activation of the plantar flexors and evertors of the ankle [41] and an increase of vestibular and proprioceptive inputs to partially compensate for the lack of visual information [21, 28, 42]. However, proprioceptive compensatory mechanisms alone do not appear to be adequate. Ozdemir et al. [42] observed that visually impaired subjects with high proprioceptive acuity had worse postural control performance than sighted people. The increase of the plantar pressures in M1 and M2 zones when the visually impaired participants had their eyes closed could be interpreted as a compensatory response that would to augment proprioceptive input to avoid loss of balance control. The plantar pressures' change seems to be an anticipatory adjustment before the changes in balance control.

**Table 4. Comparison of the stabilometric parameters between control group and visually impaired group according to etiology of the impairment.**

| | | Legally blind group | | |
|---|---|---|---|---|
| Variable | Control group | Congenital | Acquired | p-value |
| *Open eye condition* | | | | |
| $COP_{distance}$ | 327.5 (103.8) | 345.8 (140.1) | 309.3 (75.2) | 0.93 |
| $COP_{área}$ | 87.7 (82.1) | 69.03 (63.5) | 66.82 (112.4) | 0.84 |
| *Closed eye condition* | | | | |
| $COP_{distance}$ | 351.3 (57) | 336.6 (165.1) | 317.7 (50.3) | 0.53 |
| $COP_{area}$ | 110.6 (118.6) | 66.59 (102.91) | 92.04 (120.1) | 0.43 |

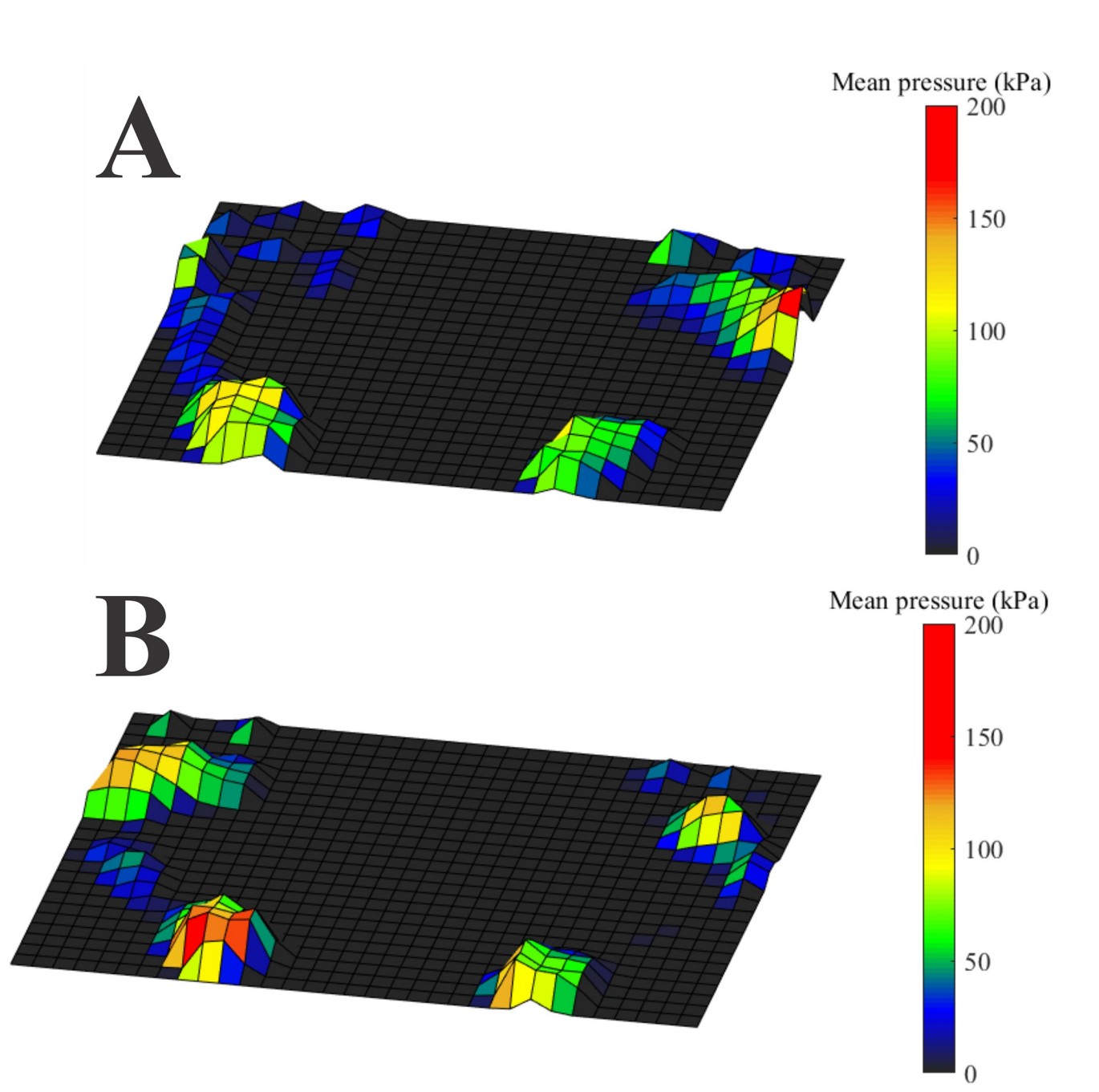

**Fig 3. Heatmap of the plantar pressure measurements obtained from the participants whose balance control data are shown in Fig 2.** (A) Sighted participant. The pressures are distributed from the hallux to mid-lateral foot and in the heel. (B) Visually impaired participant. The pressure distribution occurs mainly in the hallux, all the metatarsal region of the left foot and in the heel.

Finally, the role of hapic cues in the foot must be considered in our subjects, since it was previously demonstrated less frequent head displacements in sighted subjects than in visually impaired subjects, in experiments using haptic cues derived from a cane's contact [21].

**Table 5. Comparison of $P_{mean}$ and $P_{max}$ measured in the different foot regions of control and visually impaired participants.**

| | $P_{mean}$ | | | | | |
|---|---|---|---|---|---|---|
| | **Open eye condition** | | | **Closed eye condition** | | |
| **Region** | **CG** | **LBG** | **p-value** | **CG** | **LBG** | **p-value** |
| T1 | 6.83 (13.2) | 10.92 (13.8) | 0.27 | 8.83 (14.4) | 14.33 (13.5) | 0.67 |
| T2-5 | 3.17 (4.9) | 5.16 (4.5) | 0.12 | 3.75 (6.3) | 5.58 (6.1) | 0.32 |
| M1 | **18.25 (9.2)** | **25.75 (16.8)** | **0.01** | **18.08 (6)** | **23.5 (14.4)** | **0.005** |
| M2 | 27.08 (21.5) | 33.75 (21.3) | 0.12 | 27.33 (20.3) | 34.25 (21.7) | 0.06 |
| M3 | 40.75 (21.7) | 43.25 (19.6) | 0.41 | 35.92 (19.9) | 45.75 (24.9) | 0.15 |
| M4 | 43.92 (26.8) | 48 (19.6) | 0.44 | 39.5 (28) | 46.5 (20.4) | 0.87 |
| M5 | 24 (24.7) | 29.42 (13.6) | 0.99 | 26.17 (22.5) | 26.25 (13.8) | 0.56 |
| MF | 11.58 (11) | 13.83 (9.1) | 0.12 | 12.33 (11.6) | 14.83 (8.3) | 0.3 |
| MH | 63.92 (18) | 59.17 (18) | 0.5 | 60.92 (22) | 60.67 (22.2) | 0.8 |
| LH | 63.17 (10.1) | 57.83 (13.3) | 0.36 | 59.42 (15.8) | 56.17 (19.7) | 0.83 |
| | $P_{max}$ | | | | | |
| | **Open eye condition** | | | **Closed eye condition** | | |
| **Region** | **CG** | **LBG** | **p-value** | **CG** | **LBG** | **p-value** |
| T1 | 16.5 (43.8) | 30 (44.4) | 0.19 | 22.08 (51.8) | 44.5 (41.1) | 0.63 |
| T2-5 | 6.5 (14.2) | 10.92 (17.3) | 0.2 | 7.67 (17.5) | 14.25 (20.6) | 0.3 |
| M1 | 49.17 (29.8) | 60.83 (41.3) | 0.06 | **43.5 (19.7)** | **64 (44.8)** | **0.02** |
| M2 | **62.25 (38.6)** | **81.33 (40.5)** | **0.03** | **58.83 (35.5)** | **82.17 (37)** | **0.02** |
| M3 | 85.75 (46.7) | 89.25 (38.1) | 0.56 | 73.33 (40.4) | 90.17 (42.8) | 0.37 |
| M4 | 87 (58.2) | 90.92 (41.6) | 0.94 | 79.08 (52.1) | 86.83 (44.6) | 0.97 |
| M5 | 69.75 (67.9) | 82 (37) | 0.94 | 74.92 (62.8) | 72.42 (38.9) | 0.75 |
| MF | 36.17 (36.2) | 39.25 (12.5) | 0.59 | 41.92 (42.1) | 38.5 (18.3) | 0.63 |
| MH | 153.3 (38.8) | 145.3 (36.5) | 0.37 | 148.4 (39.2) | 153.8 (55.6) | 0.97 |
| LH | 149 (33.5) | 141.9 (30.5) | 0.41 | 140.8 (40.9) | 146.9 (53.1) | 0.96 |

CG: Control group; LBG: Legally blind group. Comparisons in bold represent significant differences using a Mann-Whitney test.

**Table 6. Comparison of the plantar pressures among controls, congenital and acquired visually impaired participants during eye aperture conditions.**

| | $P_{mean}$ | | | | | |
|---|---|---|---|---|---|---|
| | **OPEN-EYE CONDITION** | | | **CLOSED-EYE CONDITION** | | |
| | | **Legally blind group** | | | **Legally blind group** | |
| **Region** | **Control group** | **Congenital** | **Acquired** | **Control group** | **Congenital** | **Acquired** |
| T1 | 6.83 (13.2) | 9.25 (13.9) | 13.5 (10.6) | 8.83 (14.4) | 8.58 (15) | 15.67 (9.8) |
| T2-5 | 3.17 (4.9) | 4.25 (3) | 7.08 (8) | 3.75 (6.33) | 4.33 (7.2) | 7.08 (4.7) |
| M1 | 18.25 (9.2) | 25.75 (16) | 25.08 (20.2) | 18.08 (6) | 23.08 (12.2) | **24.17 (15.8)**[*] |
| M2 | 27.08 (21.5) | 33.67 (18.2) | 34.83 (25.3) | 27.33 (20.3) | 34.25 (18) | 36.92 (23.1) |
| M3 | 40.75 (21.7) | 43.25 (23.9) | 42 (23.7) | 35.92 (19.9) | 40.67 (25.08) | 49.08 (29.6) |
| M4 | 43.92 (26.8) | 46.83 (25.5) | 47.67 (24.6) | 39.5 (28) | 42.42 (25) | 48.67 (21.8) |
| M5 | 24 (24.7) | 30.17 (10.9) | 27.25 (18) | 26.17 (22.5) | 26.42 (19.5) | 25.58 (11.8) |
| MF | 11.58 (11) | 13.5 (9.9) | 14.33 (10.3) | 12.3 (11.5) | 12.3 (18.4) | 16.42 (4.9) |
| MH | 63.92 (18) | 59.42 (19.4) | 57.83 (18.8) | 60.92 (21.7) | 61.17 (24.8) | 58.67 (19.2) |
| LH | 60.75 (11.7) | 57.83 (14.6) | 57.83 (20.5) | 59.42 (15.8) | 57.67 (26.1) | 53.92 (16.3) |

*(Continued)*

**Table 6.** (Continued)

| | P$_{max}$ | | | | | |
| | OPEN-EYE CONDITION | | | CLOSED-EYE CONDITION | | |
| | | Legally blind group | | | Legally blind group | |
| Region | Control group | Congenital | Acquired | Control group | Congenital | Acquired |
|---|---|---|---|---|---|---|
| T1 | 17.58 (57.5) | 23.83 (44.7) | 43 (35.2) | 22.08 (51.8) | 25.42 (48.3) | 48.92 (27.5) |
| T2-5 | 6.5 (14.2) | 8.83 (9.4) | 18.75 (32.8) | 7.67 (17.5) | 8.58 (19) | 20.67 (19.8) |
| M1 | 49.17 (29.8) | 63.92 (44.4) | 65.83 (58.3) | 43.5 (19.7) | 59.25 (29.6) | 70.33 (52.5) |
| M2 | 62.25 (38.7) | 81.42 (36.3) | 78.83 (43.7) | 58.83 (35.5) | 75.25 (35.3) | **96.58 (52.5)**[**] |
| M3 | 85.75 (46.7) | 90.42 (44.6) | 85.25 (46.7) | 73.33 (40.4) | 79.42 (45) | 96.58 (52.5) |
| M4 | 87 (58.2) | 87.67 (47.4) | 96 (49.6) | 79.08 (52.1) | 82.5 (47) | 94.25 (57.2) |
| M5 | 69.75 (67.9) | 82 (37.7) | 78.58 (49.7) | 74.92 (62.7) | 71.67 (46.5) | 74.67 (34.8) |
| MF | 36.17 (36.2) | 41.25 (24) | 46.75 (25.8) | 41.92 (42.1) | 36.33 (47.9) | 43.17 (17.2) |
| MH | 153.3 (38.8) | 155.7 (33.8) | 134.3 (52.9) | 148.4 (39.2) | 158.4 (58.1) | 139.8 (45.8) |
| LH | 149 (33.5) | 150.5 (31.7) | 135.9 (50.5) | 140.8 (40.9) | 150.3 (52.3) | 135.2 (45.2) |

Comparisons in bold represent significant differences using Kruskal-Wallis test followed by Dunn's post hoc test.

*Higher than the control group (Dunn's test, p = 0.02).

**Higher than the control group (Dunn's test, p = 0.04).

The present study has limitations that must be considered. The sample of participants with acquired blindness is heterogeneous in terms of duration of visual loss, for example. We would hypothesize that a longer period of visual loss could increase the chances of developing compensatory mechanisms for vision loss. A comparison of the balance control in subjects with short- and long-term acquired visual loss could help clarify the role that duration of acquired visual loss might have in the development of compensatory mechanisms in the nervous system.

We concluded that acquired blindness alters the balance control mechanisms more the congenital blindness, as evidenced by the anterior displacement of the foot pressures observed in the acquired blind subjects. Development of more complete compensatory mechanisms in congenital impairment (vs acquired) are suggested as a possible explanation to the observed difference. Our findings suggest that acquired blind people need more attention concerning the risk of fall.

## Supporting information

**S1 Database.**
(XLSX)

## Acknowledgments

This research received funding from the Amazon Paraense Foundation of Studies (FAPESPA, No. 2019/589349) and the Research Funding and the National Council of Research Development (CNPq/Brazil, No. 431748/2016-0). GS was CNPq Productivity Fellow (No. 310845/2018-1).

## Author Contributions

**Conceptualization:** Anselmo Athayde Costa e Silva, Bianca Callegari, Givago Silva Souza.

**Data curation:** Anselmo Athayde Costa e Silva, Bianca Callegari, Givago Silva Souza.

**Formal analysis:** Ketlin Jaquelline Santana Castro, Railson Cruz Salomão, Newton Quintino Feitosa, Jr., Leonardo Dutra Henriques, Ana Francisca Rozin Kleiner, Anderson Belgamo, André Santos Cabral, Bianca Callegari.

**Funding acquisition:** Anselmo Athayde Costa e Silva, Givago Silva Souza.

**Investigation:** Ketlin Jaquelline Santana Castro, Railson Cruz Salomão, Newton Quintino Feitosa, Jr., Bianca Callegari.

**Methodology:** Ketlin Jaquelline Santana Castro, Railson Cruz Salomão, Newton Quintino Feitosa, Jr., Ana Francisca Rozin Kleiner, Anselmo Athayde Costa e Silva, Bianca Callegari.

**Project administration:** Bianca Callegari, Givago Silva Souza.

**Supervision:** Bianca Callegari, Givago Silva Souza.

**Visualization:** Givago Silva Souza.

**Writing – original draft:** Ketlin Jaquelline Santana Castro, Leonardo Dutra Henriques, Ana Francisca Rozin Kleiner, Anderson Belgamo, André Santos Cabral, Anselmo Athayde Costa e Silva, Bianca Callegari, Givago Silva Souza.

**Writing – review & editing:** Ketlin Jaquelline Santana Castro, Railson Cruz Salomão, Newton Quintino Feitosa, Jr., Leonardo Dutra Henriques, Ana Francisca Rozin Kleiner, Anderson Belgamo, André Santos Cabral, Anselmo Athayde Costa e Silva, Bianca Callegari, Givago Silva Souza.

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
