## [Decision Letter · Decision Letter 0]

11 Dec 2020

PONE-D-20-33643

CHANGES IN PLANTAR LOAD DISTRIBUTION IN VISUALLY-IMPAIRED SUBJECTS

PLOS ONE

Dear Dr. Souza,

Thank you for submitting your manuscript to PLOS ONE. After careful consideration, we feel that it has merit but does not fully meet PLOS ONE’s publication criteria as it currently stands. Therefore, we invite you to submit a revised version of the manuscript that addresses the points raised during the review process.

Two experts in the field have carefully evaluated the manuscript entitled, "CHANGES IN PLANTAR LOAD DISTRIBUTION IN VISUALLY-IMPAIRED SUBJECTS". Their comments are appended below.

The first reviewer gave rather favorable comments but not satisfactory. The second referee pointed out the drawbacks need to be revised from all the aspect of the manuscript.

This Academic Editor advise the authors to be consulted with a professional English Editing service before submission.

We look forward to receiving your revised manuscript.

Kind regards,

Manabu Sakakibara, Ph.D.

Academic Editor

PLOS ONE

Journal Requirements:

Reviewers' comments:

Reviewer's Responses to Questions

**Comments to the Author**

1. Is the manuscript technically sound, and do the data support the conclusions?

Reviewer #1: Partly

Reviewer #2: Partly

2. Has the statistical analysis been performed appropriately and rigorously? 

Reviewer #1: Yes

Reviewer #2: No

3. Have the authors made all data underlying the findings in their manuscript fully available?

Reviewer #1: Yes

Reviewer #2: Yes

4. Is the manuscript presented in an intelligible fashion and written in standard English?

Reviewer #1: Yes

Reviewer #2: No

5. Review Comments to the Author

Reviewer #1: Thank you for the opportunity to review the manuscript titled "CHANGES IN PLANTAR LOAD DISTRIBUTION IN VISUALLY-IMPAIRED SUBJECTS" (PONE-D-20-33643). This is an interesting descriptive study on the consequences of visual deprivation on balance and plantar pressures.

The authors provide a brief but interesting introduction, but do not include all the current scientific evidence on the subject.

Below, I suggest a series of changes to improve the quality of the manuscript, which I beg you to take as constructive criticism:

1. It is easier to make changes and indicate specific errors if the text has numbered lines. It is a suggestion for future submits.

2. In the abstract, it is no longer clear from the beginning whether we are talking about "visually impairment subjects" or "blind people", since as the authors indicate at the end of the discussion, "blind subjects" is not equal to "visual impairment ". It is better to use the same word mark throughout the text.

3. There are recent studies on plantar pressure and balance in healthy subjects that may help to support certain results of this study. For example:

Sánchez-González, María Carmen et al. Visual Binocular Disorders and Their Relationship with Baropodometric Parameters: A Cross-Association Study. BioMed Research International Volume 2020, Article ID 6834591, 9 pages. DOI: 10.1155 / 2020/6834591. It would be nice to include recent articles like this to improve the introduction and discussion.

4. In the methodology, the selection criteria of the sample have not been described and the characteristics of the sample are not exactly detailed until Table 1 and 2. Where is the sample collected from? It is too striking that the sample is so homogeneous in relation to age and body mass index.

5. It talks about the two situations in which the "visually impairment" subjects are measured: with eyes open and closed, but, if they were blind, as you say later, why have these two situations been taken into account?

6. Improve the quality of Figure 1, it looks a little blurry. It would be interesting to explain what the different zones consist of in the same text and not in the figure legends.

7. It is also striking that the sample is divided almost in half into "acquired" and "congenital" visually impairment subjects. Was it done randomly or was it sampled of some kind to select them?

8. The ages at which vision was lost in the "acquired" group are very different (30 years vs 2 years), could this not affect the previous visual experience and therefore the results?

9. Figure 2, like Figure 1, should be explained in text to reduce the figure legend.

10. Could tables 6 and 7 be unified so that the results of P mean and P max can be equated?

11. The discussion must be improved, including references that can justify the results, as well as not generalizing the results (as in the first section) where it literally says "the visually impaired participants had higher pressure in the first and second metatarsals". This has not been the case in all cases, it would be necessary to specify and try to justify these findings. Be careful with generalizing as the sample is not too large.

12. Merge the references in the discussion.

13. The last paragraph that explains the blindness, should be explained much earlier in the text of the manuscript, including putting it in the sample and its selection.

Reviewer #2: In this study authors aimed to investigate the effects of visual impairment on postural stability, considering both the center of pressure outcomes and the plantar surface distribution. The paper is interesting but results have poor potential broader relevance, since most of them were not significant. Furthermore, there are several issues which will require your attention.

1. The English in the present manuscript is not of publication quality and requires an improvement. Please carefully proof-read spell check to eliminate grammatical errors. Some periods need also to be revised, like for example the following, in the “Abstract” section:

“We aimed to compare the center of pressure and pressure in the feet plantar surface measured by sighted and visually impaired subjects”

Or the following in the “Introduction” section:

“Visual, vestibular, and proprioceptive inputs inform those forces to the brain in order to reach the equilibrium status on a standing position”.

2. Most of the references used are older than five years. Authors should focus on recent papers, like the following ones:

“Caldani, S., Bucci, M. P., Tisné, M., Audo, I., Van Den Abbeele, T., & Wiener-Vacher, S. (2019). Postural instability in subjects with usher syndrome. Frontiers in Neurology, 10.”

“D'Antonio, E., Tieri, G., Patané, F., Morone, G., & Iosa, M. (2020). Stable or able? Effect of virtual reality stimulation on static balance of post-stroke patients and healthy subjects. Human movement science, 70, 102569.”

Alghadir, A. H., Alotaibi, A. Z., & Iqbal, Z. A. (2019). Postural stability in people with visual impairment. Brain and Behavior, 9(11), e01436.

3. The Introduction is not complete. It should establish the context of the research by summarizing current understanding and background information about the topic, stating the purpose of the work, briefly explaining the rationale and the methodological approach, and highlighting the potential outcomes your study can reveal. Authors should describe previous studies and report the main results obtained from them. On the basis of these they should formulate their hypotheses.

4. Methods: Authors should describe in details the inclusion/exclusion criteria considering also the motor impairments.

5. Methods: Authors wrote that the noise was avoided in the room during recordings. Please explain this point, adding details about the methodology applied in order to avoid noise.

6. Methods: How did the authors consider the differences between the kind of visual impairment?

7. Figure 1: The figure is not clear. Please add some details in order to clarify that the shape is depicting a foot. Authors should also add a legend of the different areas they are representing. The resolution of the figure should also be improved.

8. Figure 2: Authors should change the scale of the figure in order to clearly show the differences between the four cases represented. What do authors want to highlight with this figure? Please add some details in order to clarify this point.

9. Results: Authors should explain how they applied the reduction of the p-value in the post-hoc analysis in order to appropriately perform the statistical analysis.

10. Results: Most of the results are not significant. What does it mean?

11. The limitations of the study should be integrated in the paper.

12. The conclusion should be also integrated in the paper.

6. PLOS authors have the option to publish the peer review history of their article (what does this mean?). If published, this will include your full peer review and any attached files.

Reviewer #1: No

Reviewer #2: **Yes: **Erika D'Antonio

---

## [Author Response · Author response to Decision Letter 0]

3 Mar 2021

Reviewer's Responses to Questions

Comments to the Author

1. Is the manuscript technically sound, and do the data support the conclusions?

Reviewer #1: Partly

Reviewer #2: Partly

A. We have made substantial modifications in the manuscript addressing the reviewers’ suggestions. We hope that the new version can be clearer for the readers.

2. Has the statistical analysis been performed appropriately and rigorously?

Reviewer #1: Yes

Reviewer #2: No

A. We added the suggestions for the statistics description.

3. Have the authors made all data underlying the findings in their manuscript fully available?

Reviewer #1: Yes

Reviewer #2: Yes

A. Thanks.

4. Is the manuscript presented in an intelligible fashion and written in standard English?

Reviewer #1: Yes

Reviewer #2: No

A. A native English-speaker ed0ited the final version of the present manuscript.

5. Review Comments to the Author

Reviewer #1: Thank you for the opportunity to review the manuscript titled "CHANGES IN PLANTAR LOAD DISTRIBUTION IN VISUALLY-IMPAIRED SUBJECTS" (PONE-D-20-33643). This is an interesting descriptive study on the consequences of visual deprivation on balance and plantar pressures.

The authors provide a brief but interesting introduction, but do not include all the current scientific evidence on the subject.

Below, I suggest a series of changes to improve the quality of the manuscript, which I beg you to take as constructive criticism:

1. It is easier to make changes and indicate specific errors if the text has numbered lines. It is a suggestion for future submits.

A. Thanks. We included the numbered lines in the new version.

2. In the abstract, it is no longer clear from the beginning whether we are talking about "visually impairment subjects" or "blind people", since as the authors indicate at the end of the discussion, "blind subjects" is not equal to "visual impairment ". It is better to use the same word mark throughout the text.

A. In the new version of the manuscript we referred to the participants of our study as legally blind subjects, while we used other qualification (such as visually impaired) when is referenced participants from other investigations as is described by them.

3. There are recent studies on plantar pressure and balance in healthy subjects that may help to support certain results of this study. For example:

Sánchez-González, María Carmen et al. Visual Binocular Disorders and Their Relationship with Baropodometric Parameters: A Cross-Association Study. BioMed Research International Volume 2020, Article ID 6834591, 9 pages. DOI: 10.1155 / 2020/6834591. It would be nice to include recent articles like this to improve the introduction and discussion.

A. Thanks for the indication. We added it to the manuscript as well as other recent articles.

4. In the methodology, the selection criteria of the sample have not been described and the characteristics of the sample are not exactly detailed until Table 1 and 2. Where is the sample collected from? It is too striking that the sample is so homogeneous in relation to age and body mass index.

A. We added more information about the sample in the Methods section as follows (Page 7, Line 136):

“Both groups were age-, body mass index- (BMI), and male/female proportion matched. The sample was homogeneous to age, BMI, and male/female proportion.”

5. It talks about the two situations in which the "visually impairment" subjects are measured: with eyes open and closed, but, if they were blind, as you say later, why have these two situations been taken into account?

A. We added sentences about it in the Methods and Discussion sections.

In Methods (Page 5, Line 110):

“All blind subjects that participated in the present study were legally blind (i.e., visual acuity was equal to or worse than 20/200) (Table 1). The definition of blindness is based on foveal vision (central vision), and most of the blind participants had some luminance perception, indicating some peripheral visual function. It is unclear the role of central or peripheral vision in balance control, if both have equal importance, or if both have complementary functionality [31-35].”

In Discussion section (Page 16, Line 269):

“Because we found that differences between the acquired and congenital blind groups in plantar pressures occurred mainly in the closed-eye condition, residual light perception (present in most of the acquired blind group) is a likely an important factor that assists in performance during the balance control task.

In addition, the visual system has more than one pathway to the brain; and conscious visual perception is generated in only one of these pathways (from retina to primary visual cortex, V1). However, other visual pathways could be contributing to the balance control even in visually impaired or blind subjects (for example from retina to superior colliculus or pulvinar nucleus), as has been observed in cases of so-called “blindsight” demonstrating residual vision in several visual diseases [39-41].”

6. Improve the quality of Figure 1, it looks a little blurry. It would be interesting to explain what the different zones consist of in the same text and not in the figure legends.

A. Done.

7. It is also striking that the sample is divided almost in half into "acquired" and "congenital" visually impairment subjects. Was it done randomly or was it sampled of some kind to select them?

A. The sample of legally blind subjects was recruited from the José Alvares de Azevedo school for the blind and visually impaired. Participants were not compensated financially. An ophthalmologist evaluated all blind participants and classified them as acquired or congenital blinds. We evaluated all the available students to include the large possible number of subjects in the sample. For luck, we had similar number of participants in both groups.

8. The ages at which vision was lost in the "acquired" group are very different (30 years vs 2 years), could this not affect the previous visual experience and therefore the results?

A. Thanks for the question. It is not possible to assert about it, but we consider that represents a limitation of the study. We added this observation in the Discussion section as follows (Page 15, Line 256).

“Because of the similarity in COP measures obtained from participants with congenital or acquired blindness, a working hypothesis could be that the mean duration of acquired impairment (11 ± 9.2 years) was adequate for the development and consolidation of compensatory balance-control mechanisms.

However, against such a hypothesis, the presence of significant differences in metatarsal head pressures argues for an incomplete establishment of these mechanisms in the participants with acquired blindness. Congenitally blind subjects would have an opportunity to develop compensatory mechanisms during their development in early childhood, while the ability to develop compensatory mechanisms in acquired visual loss would depend on many factors, including age of onset relative to any possible critical periods, and duration of the visual impairment. Moreover, it is possible that qualitatively different compensatory mechanisms for balance control might be utilized in acquired vs congenital visual impairment.”

9. Figure 2, like Figure 1, should be explained in text to reduce the figure legend.

A. Done (Page 9, Line 188):

“Typical statokinesiograms were found for all participants, in which anteriorposterior displacements are larger than the laterolateral displacements during the test duration. Figure 2 shows representative statokinesiograms obtained from one representative participant in each group, and no systematic or qualitative differences between the groups could be observed between the statokinesiograms of these participants.”

10. Could tables 6 and 7 be unified so that the results of P mean and P max can be equated?

A. Done.

11. The discussion must be improved, including references that can justify the results, as well as not generalizing the results (as in the first section) where it literally says "the visually impaired participants had higher pressure in the first and second metatarsals". This has not been the case in all cases, it would be necessary to specify and try to justify these findings. Be careful with generalizing as the sample is not too large.

A. Thanks for the suggestions. We rewrote the Discussion following the suggestion.

12. Merge the references in the discussion.

A. Done.

13. The last paragraph that explains the blindness, should be explained much earlier in the text of the manuscript, including putting it in the sample and its selection.

A. Thanks. We replaced to the Methods section (Page 5, Line 110).

Reviewer #2: In this study authors aimed to investigate the effects of visual impairment on postural stability, considering both the center of pressure outcomes and the plantar surface distribution. The paper is interesting but results have poor potential broader relevance, since most of them were not significant. Furthermore, there are several issues which will require your attention.

1. The English in the present manuscript is not of publication quality and requires an improvement. Please carefully proof-read spell check to eliminate grammatical errors. Some periods need also to be revised, like for example the following, in the “Abstract” section:

“We aimed to compare the center of pressure and pressure in the feet plantar surface measured by sighted and visually impaired subjects”

Or the following in the “Introduction” section:

“Visual, vestibular, and proprioceptive inputs inform those forces to the brain in order to reach the equilibrium status on a standing position”.

A. The new version of the manuscript was edited by an English native-speaker and we hope to be suitable for publication quality. 

2. Most of the references used are older than five years. Authors should focus on recent papers, like the following ones:

“Caldani, S., Bucci, M. P., Tisné, M., Audo, I., Van Den Abbeele, T., & Wiener-Vacher, S. (2019). Postural instability in subjects with usher syndrome. Frontiers in Neurology, 10.”

“D'Antonio, E., Tieri, G., Patané, F., Morone, G., & Iosa, M. (2020). Stable or able? Effect of virtual reality stimulation on static balance of post-stroke patients and healthy subjects. Human movement science, 70, 102569.”

Alghadir, A. H., Alotaibi, A. Z., & Iqbal, Z. A. (2019). Postural stability in people with visual impairment. Brain and Behavior, 9(11), e01436.

A. Thanks for the indications. We added them to the manuscript.

3. The Introduction is not complete. It should establish the context of the research by summarizing current understanding and background information about the topic, stating the purpose of the work, briefly explaining the rationale and the methodological approach, and highlighting the potential outcomes your study can reveal. Authors should describe previous studies and report the main results obtained from them. On the basis of these they should formulate their hypotheses.

A. We rewrote the Introduction section following the suggestions.

4. Methods: Authors should describe in details the inclusion/exclusion criteria considering also the motor impairments.

A. We added more information about the inclusion/exclusion criteria in the Methods section (Page 5, Line 99).

5. Methods: Authors wrote that the noise was avoided in the room during recordings. Please explain this point, adding details about the methodology applied in order to avoid noise.

A. We changed the sentence. We wrote that no conversation was not allowed during the recording sessions in the Methods section (Page 8, Line 152) as follows:

“Conversation was not allowed during the recording sessions, except for the orientations of the participants by the examiners.”

6. Methods: How did the authors consider the differences between the kind of visual impairment?

A. We added a sentence about the differences between the kind of visual impairment in the Methods section as follows (Page 5, Line 108):

“After the ophthalmological examination and the history of the present illness we divided the sample into acquired and congenital blind participants.”

7. Figure 1: The figure is not clear. Please add some details in order to clarify that the shape is depicting a foot. Authors should also add a legend of the different areas they are representing. The resolution of the figure should also be improved.

A. We created a new Figure 1 following the suggestion.

8. Figure 2: Authors should change the scale of the figure in order to clearly show the differences between the four cases represented. What do authors want to highlight with this figure? Please add some details in order to clarify this point.

A. Done (Page 9, Line 188):

“Typical statokinesiograms were found for all participants, in which anteriorposterior displacements are larger than the laterolateral displacements during the test duration. Figure 2 shows representative statokinesiograms obtained from one representative participant in each group, and no systematic or qualitative differences between the groups could be observed between the statokinesiograms of these participants.”

Typical statikinesiograms were found for all participants, which anteriorposterior displacements are larger than the laterolateral displacements during the test duration. Figure 2 shows representative statokinesiograms obtained from one participant in each group, and no systematic or qualitative differences between the groups could be observed.

9. Results: Authors should explain how they applied the reduction of the p-value in the post-hoc analysis in order to appropriately perform the statistical analysis.

A. We added a sentence with explanation about the adjusted p-values as follows (Page 9, Line 180):

“We calculated the adjusted p-values for multiplicity using the Bonferroni correction.”

10. Results: Most of the results are not significant. What does it mean?

A. We included in the Discussion a sentence debating about the non-significant and significant results as follows (Page 15, Line 249): 

“Additionally, we also observed that most of the foot zones of the legally blind participants had no significant pressure differences compared to controls, except for the higher maximum pressures in the first and second metatarsal heads (M1 and M2 zones) in patients in the closed-eye condition. These differences could be attributed to the contribution of the acquired blind participants' data [15]. The small number of significant differences indicates that mild disturbances in the motor control of the balance are present and maybe can be ignored in conventional evaluation of the balance.”

11. The limitations of the study should be integrated in the paper.

A. Done (Page 17, Line 296):

“The present study has limitations that must be considered. The sample of participants with acquired blindness is heterogeneous in terms of duration of visual loss, for example. We would hypothesize that a longer period of visual loss could increase the chances of developing compensatory mechanisms for vision loss. A comparison of the balance control in subjects with short- and long-term acquired visual loss could help clarify the role that duration of acquired visual loss might have in the development of compensatory mechanisms in the nervous system.”

12. The conclusion should be also integrated in the paper.

A. Done (Page 17, Line 303):

“We concluded that acquired blindness alters the balance control mechanisms more the congenital blindness, as evidenced by the anterior displacement of the foot pressures observed in the acquired blind subjects. Development of more complete compensatory mechanisms in congenital impairment (vs acquired) are suggested as a possible explanation to the observed difference. Our findings suggest that acquired blind people need more attention concerning the risk of fall.”

6. PLOS authors have the option to publish the peer review history of their article (what does this mean?). If published, this will include your full peer review and any attached files.

Do you want your identity to be public for this peer review? For information about this choice, including consent withdrawal, please see our Privacy Policy.

Reviewer #1: No

Reviewer #2: Yes: Erika D'Antonio

A. Thanks for both reviewers for the extensive evaluation. We hope that the new version has been improved for publication.

---

## [Decision Letter · Decision Letter 1]

19 Mar 2021

CHANGES IN PLANTAR LOAD DISTRIBUTION IN LEGALLY BLIND SUBJECTS

PONE-D-20-33643R1

Dear Dr. Souza,

We’re pleased to inform you that your manuscript has been judged scientifically suitable for publication and will be formally accepted for publication once it meets all outstanding technical requirements.

Kind regards,

Manabu Sakakibara, Ph.D.

Academic Editor

PLOS ONE

Additional Editor Comments (optional):

Reviewers' comments:

Reviewer's Responses to Questions

**Comments to the Author**

1. If the authors have adequately addressed your comments raised in a previous round of review and you feel that this manuscript is now acceptable for publication, you may indicate that here to bypass the “Comments to the Author” section, enter your conflict of interest statement in the “Confidential to Editor” section, and submit your "Accept" recommendation.

Reviewer #1: All comments have been addressed

Reviewer #2: All comments have been addressed

2. Is the manuscript technically sound, and do the data support the conclusions?

Reviewer #1: Yes

Reviewer #2: Yes

3. Has the statistical analysis been performed appropriately and rigorously? 

Reviewer #1: Yes

Reviewer #2: Yes

4. Have the authors made all data underlying the findings in their manuscript fully available?

Reviewer #1: Yes

Reviewer #2: Yes

5. Is the manuscript presented in an intelligible fashion and written in standard English?

Reviewer #1: Yes

Reviewer #2: Yes

6. Review Comments to the Author

Reviewer #1: The authors have incorporated all the suggestions made, improving the content and the methodology. Now the manuscript is much more enriched and improved.

Reviewer #2: Authors have adequately addressed all the comments raised in a previous round of review and the manuscript is now acceptable for publication.

7. PLOS authors have the option to publish the peer review history of their article (what does this mean?). If published, this will include your full peer review and any attached files.

Reviewer #1: No

Reviewer #2: **Yes: **Erika D'Antonio

---

## [Editor Report · Acceptance letter]

5 Apr 2021

PONE-D-20-33643R1 

CHANGES IN PLANTAR LOAD DISTRIBUTION IN LEGALLY BLIND SUBJECTS 

Dear Dr. Souza:

I'm pleased to inform you that your manuscript has been deemed suitable for publication in PLOS ONE. Congratulations! Your manuscript is now with our production department. 

Kind regards, 

on behalf of

Dr. Manabu Sakakibara 

Academic Editor

PLOS ONE